# An improved pixel-based water vapor tomography model

**Yibin Yao [1, 2, *], Linyang Xin [1] and Qingzhi Zhao [3]**

[1]  School of Geodesy and Geomatics, Wuhan University, Wuhan 430079, China;
    ybyao@whu.edu.cn (Y.Y); linyangxin@whu.edu.cn (L.X);
[2]  Key Laboratory of Geospace Environment and Geodesy, Ministry of Education, Wuhan University, Wuhan 430079, China
[3]  College of Geomatics, Xi'an University of Science and Technology, Xi'an 710054, China;
    zhaoqingzhia@163.com
*    Correspondence: ybyao@whu.edu.cn; Tel.: +86-027-68758401

**Abstract**: As a new detection method of three-dimensional water vapor, the ground-based water vapor tomography technique using Global Navigation Satellite Systems (GNSS) observations can obtain the high spatial and temporal distribution information of tropospheric water vapor. Since the troposphere tomography was proposed, most previous studies belong to the pixel-based method, dividing the interest area into three-dimensional voxels of which the water vapor density of each voxel center is taken as the average water vapor density. However, the abovementioned method can only find the water vapor density value of the center of each voxel, which is unable to express the continuous change of water vapor in space and destroys the spatial continuity of the water vapor variation. Moreover, when using the pixel-based method, too many voxels are needed to express the water vapor density, which leads to the problem of too many coefficients to be estimated. After analyzing the limitations of the traditional pixel-based tropospheric tomography technique, this paper proposes an improved pixel-based GNSS tropospheric water vapor tomography model. The tomographic experiments were validated using the data from 11 GNSS stations from the Hong Kong Satellite Positioning Reference Station Network (SatRef) collected between 25 March and 25 April 2014. The comparison between tomographic results and the European Centre for Medium-Range Weather Forecasts (ECMWF) data is mainly used to analyze the accuracy of the new improved model under different conditions, for showing that this improved model is superior to the traditional pixel-based model in terms of root-mean-square error (RMSE) and bias. The tomography water vapor profiles of the improved model were also evaluated using radiosonde data to show the efficiency of the proposed model. Results show that the new model has more advantages than the traditional pixel-based model on the RMSE, especially when obtaining the water vapor in voxels without the penetration of GNSS rays, which is improved by 5.88%. This improved model also solves the aforesaid limitations with more ease and convenience in expression.

**Keywords:** GNSS; water vapor tomography; ECMWF; Radiosonde

## 1. Introduction

The distribution of water vapor is complex and highly variable, and water vapor, as the most active key component of the atmosphere, is indeed hard to describe accurately (Rocken et al., 1997). An in-depth understanding of temporal and spatial variation of water vapor plays an important role in improving the accuracy of weather forecasting and early warning of disastrous weather (Weckwerth et al., 2004).

GNSS water vapor monitoring techniques can not only acquire the two-dimensional spatial and temporal distribution of water vapor in the horizontal direction (Bevis et al., 1994; Emardson et al., 1998; Baltink et al., 2002; Bock et al., 2005) but can also use a three-dimensional tomography method to reconstruct the vertical structure of water vapor at high temporal-spatial resolution (Flores et al., 2000; Seko et al., 2000; Macdonald et al., 2002).

Braun et al. (1999) first proposed the concept of reconstructing the tropospheric water vapor structure using 20 GPS stations in a regional observational network. Flores et al. (2000) first presented a method of recovering the slant wet delay (SWD) and obtained the water vapor density using the observation of SWD by singular value decomposition (SVD) combined with a least square method. In the same year, Hirahara (2000) used different methods to conduct tropospheric tomography experiments, which also confirmed the feasibility of obtaining three-dimensional water vapor fields using GPS technology. Since then, many scholars have studied GNSS troposphere tomography techniques and completed many research experiments (Rohm et al., 2014; Yao et al., 2016; Ding et al., 2018; Zhao et al., 2018).

Regarding the tropospheric tomography model solution and algorithm improvement, Hirahara (2000) conducted a four-dimensional tropospheric wet refractivity retrieval of the GPS network from Shigaraki and solved the observation equations using the damping least square method, which is commonly used in seismic tomography. Braun et al. (2003, 2004) overcame the sensitivity problem in tomographic results by using the extended sequential filtering method. Perler et al. (2011) presented a new parameterization method for the water vapor retrieval. The measured and simulated data proved that this method can obtain better tomographic solution results of water vapor. Nilsson and Gradinarsky (2006) obtained the tropospheric tomographic results directly from the original GNSS phase observations combined with the Kalman filter method. Rohm and Bosy (2009) used the Moore-Penrose pseudo-inverse of variance-covariance to solve the linear equations and emphasized the ill-posed tomography equation. Zhao and Yao (2017) obtained good results by using the optimal grid-making method for water vapor tomography. In the meantime, a method of using the side-penetrating signals for tomography was proposed to improve the effect of GNSS ray utilization rate. Aghajany and Amerian (2017) obtained the tomography results of water vapor profiles, applying 3D ray tracing technique based on Eikonal equations and ERA-I numerical weather prediction data to perform the signal path. Dong and Jin (2018) reconstructed the 3-D water vapor density using the combined multi-GNSS system, showing that the accuracy of GNSS tropospheric tomography results could be improved by 5% from the GPS-only system to the dual-system (GPS+GLONASS). Besides, the virtual reference station approach (Vollath et al., 2013; Marel, 1998), an effective method to attenuate the effects of atmospheric errors in long-distance dynamic positioning, could also be used in GNSS tropospheric tomography.

Although GNSS tomography techniques have been developed for more than two decades, it has been challenging to enhance the water vapor quality and the stability of the solution results through the multi-system and multi-source data combination method and improve the solution and algorithm of the tropospheric tomography model. However, in the previous studies, most water vapor tomography methods belong to the pixel-based model, which means that the three-dimensional meshes of the study area were

used, and the water vapor density at the center of each voxel was taken as the average water vapor density
of the whole voxel. Only could find the water vapor density value of the center of each voxel, the pixel-
based tomography is unable to continuously express the change of water vapor in space and also breaks the
spatial continuity of water vapor. Since the three-dimensional water vapor density is stored through the
voxels, a large amount of voxel information (the spatial position, the water vapor density within the voxel,
etc.) is required when describing the spatial water vapor density distribution, which is inconvenient for later
use (Yao et al., 2013). What's more, though some constraints could be put on apriori models in order to
overcome the ill-posed problem in the pixel-based tomography, some errors due to empirical constraints
would be added artificially. Thus, this paper analyzes the limitations of the traditional pixel-based
tropospheric tomography and proposes an improved pixel-based water vapor tomography model. This
model combines the advantages of facilitating the continuity of water vapor expression in spatial-temporal
distribution efficiently and retrieving the three-dimensional water vapor distribution in the interest region
easily. The experimental results show that the accuracy of the improved model is enhanced, and the new
model has more advantages when obtaining water vapor in voxels without GNSS rays penetrating. Under
strong rainfall weather conditions, the tomographic results of the improved model are more stable and
reliable.
**2. An improved pixel-based tropospheric tomography model**
*2.1. Establishment of the traditional tropospheric tomography model*
2.1.1. Retrieval of SWV
For tropospheric tomography, the most important observation is the slant water vapor (SWV), which
is related to the water vapor density and can be defined by
$$SWV = \int_S \rho_V \, ds \quad (1)$$

where *s* represents the path of the satellite signal ray, and $\rho_V$ is the water vapor density (units: g/m³).
SWV can be obtained by the following method:
$$SWV = \frac{10^6}{R_\omega[(k_3 / T_m) + k_2']} \cdot SWD \quad (2)$$

where $k_2'$=16.48 K hPa⁻¹, $k_3$=3.776×10⁵ K² hPa⁻¹, and $R_\omega$=461 J kg⁻¹ K⁻¹, which represent the specific
gas constants for water vapor. $T_m$ is the weighted mean tropospheric temperature, calculated from an
empirical equation proposed by Liu et al. (2001) using the meteorological measurements. SWD is the slant
wet delay, which may be given as
$$SWD_{elv,\varphi} = m_{wet}(elv) \times ZWD + m_{wet}(elv) \times \cot(elv) \times (G_{NS}^w \times \cos\varphi + G_{EW}^w \times \sin\varphi) + R \quad (3)$$

where *elv* is the satellite elevation, $\varphi$ is the azimuth, $m_{wet}$ is the wet mapping function, $G_{NS}^w$ and $G_{EW}^w$
are the wet delay gradient parameters in the north-south and east-west directions, respectively. *R* refers to
the unmodeled zero difference residuals that may involve unmodeled influence on the three-dimensional
spatial water vapor distribution, which can make up for the lack of tropospheric anisotropy using only the
gradient term (Bi et al., 2006). Since the GAMIT software only provides the double difference residuals,
the zero difference residuals in this paper are obtained from the double difference residuals according to
the method proposed by Alber et al. (2000). ZWD is the zenith wet delay, which is extracted from the zenith
tropospheric delay (ZTD) by separating the zenith hydrostatic delay (ZHD) using equation ZWD=ZTD-
ZHD. ZHD can be calculated precisely using surface pressure based on the Saastamoinen model
(Saastamoinen, 1972):
$$ZHD = \frac{0.002277 \times P_s}{1 - 0.00266 \times \cos(2\varphi) - 0.00028 \times H} \quad (4)$$

where $P_s$ is the surface pressure (unit: hPa), $\varphi$ is the latitude of the station, and $H$ is the geodetic
height (unit: km). The unit of ZHD is meter.
Since the SWV is obtained, the tomographic area can be discretized into a number of voxels, in which
the water vapor density is a constant during a given period of time. Therefore, a linear equation relating the
SWV and the water vapor density can be established as follows (Chen and Liu, 2014):

$$SWV^p = \sum_{ijk} (D_{ijk}^p \bullet \rho_{ijk}) \quad (5)$$

where $SWV^p$ is the slant water vapor of $\rho$ th signal path (unit: mm). i, j, and k are the positions of
discrete tomographic voxels in the longitudinal, latitudinal and vertical directions, respectively. $D_{ijk}^p$ is
the distance of the $\rho$ th signal in voxel (i, j, k) (unit: km). $\rho_{ijk}$ is the water vapor density in a given voxel
(i, j, k) (unit: g/m$^3$). A matrix form of this observation equation can be rewritten as follows (Flores et al.,
2000; Chen and Liu, 2014):
$$y_{m \times 1} = A_{m \times n} \bullet \rho_{n \times 1} \quad (6)$$

where $m$ is the number of total SWVs, and $n$ is the number of voxels in the tomographic area. $y$ is the
observed value here as the SWV, which penetrates the whole interest area, $A$ is the coefficient matrix of the
signal transit distances through the voxels, and $\rho$ is the column vector of the unknown water vapor
density.
2.1.2. Constraint equations of the tomography modeling
Solving for the unknown water vapor density in Eq. (6) is actually an inversion algorithm issue as the
design matrix A is a large sparse matrix, whose normal equation is singular, leading to numerical problems
when using a direct inversion method (Bender et al., 2011). To overcome this rank deficiency problem,
constraint equations are often introduced to the tomography equation (Flores et al., 2000; Troller et al.,
2002; Rohm and Bosy, 2009; Bender et al., 2011). In our study, the horizontal constraint equation is imposed
by the Gauss-weighted functional method (Guo et al., 2016) and the vertical constraint equation is imposed
by the functional relationship of the exponential distribution (Cao, 2012), respectively. The final
tomography model is then obtained as
$$\begin{pmatrix} A_{m \times n} \\ H_{m \times n} \\ V_{m \times n} \end{pmatrix} \bullet \rho_{n \times 1} = \begin{pmatrix} y_{m \times n} \\ 0_{m \times n} \\ 0_{m \times n} \end{pmatrix} \qquad (7)$$

where $H$ and $V$ are the coefficient matrices of horizontal and vertical constrains, respectively. In order to
obtain the inverse matrix shown in Eq. (7), singular value decomposition is used in this paper (Flores et al.,
150  2000).

*2.2. An improved pixel-based water vapor tomography model*
The improved tomography model proposed in this paper can take full advantage of facilitating the
continuity of water vapor expression efficiently in spatial-temporal distribution and calculating the water
vapor density conveniently. The improved tomography model begins to obtain the water vapor density
saved as the observation value from voxels penetrated by GNSS rays using the traditional pixel-based
tomography model and then obtains the optimal polynomial function of each layer through adaptive training.
Using the optimal polynomial fitting function of each layer with known coefficients, the water vapor density
can finally be calculated in any tomographic region by given the latitude, longitude and the altitude. Specific
steps are as follows:
First, use the traditional pixel-based water vapor tomography model to obtain the initial water vapor
density from voxels penetrated by GNSS rays as the observation values for obtaining the optimal
polynomial function coefficients of each layer.
Second, normalize the coordinates of each voxel center in the tomographic area. Since the polynomial
fitting of the water vapor at each tomographic layer is in essence establishing the relationship between the
latitude as well as the longitude of the tomographic region and the water vapor density. The general
expression is:
$$V_d = a_0 + a_1 B + a_2 L + a_3 BL + a_4 B^2 + a_5 L^2 + a_6 B^2 L \cdots \quad (8)$$

where $B$ is the latitude, $L$ is the longitude, and $V_d$ represents the water vapor density. Polynomial
coefficients such as $a_i$ are obtained via the least squares method. In the process of solving, because the
numerical values of the latitude and longitude are not small, the magnitude of multiple power may be larger
than $10^4$, which will lead to the ill-posed problem of the design matrix in the inversion process and affect
the reliability of the estimated coefficients. To ensure that the design matrix constructed will be relatively
stable in the inversion process, the latitude and longitude coordinates $B$ and $L$ need to be normalized. The
specific methods are as follows:
$$B^* = \frac{B - \mu_B}{\sigma_B}$$
$$L^* = \frac{L - \mu_L}{\sigma_L}$$
$$(9)$$

where $B^*$ and $L^*$ are the normalized latitude and longitude, respectively, and $B$ and $L$ are the
latitude and longitude in the initial region range. $\mu$ is the average value of the latitude or longitude, and
$\sigma$ is the standard deviation of the latitude or longitude.
Third, determine the layered optimal polynomial function of the improved tomography model through
adaptive training.

- First, based on the size of the selected tomographic region, determine the highest
  polynomial fit order. In this paper, the highest polynomial fit order chosen as 5 turns out to
  be generally sufficient.
- Through obtaining the water vapor density from voxels penetrated by GNSS signal rays in
  the tomographic region of each layer as the input value and constantly trying out new
  polynomial functions, the optimal polynomial function of each layer is obtained by
  simulated training.

  During the processes of training and comparison, the number of voxels penetrated by GNSS
  rays initially should be paid attention to since the number of estimated coefficients need to
  be less than that of the voxels penetrated by GNSS rays in each layer. Under this premise,
  the over-fitting problem should also be avoided, otherwise it would be counterproductive.
- Finally, after the comparison of training results of multi-group polynomial functions at
  different levels, the polynomial function with the minimum RMSE value obtained from the
  water vapor density of the post-fitting layer and that of the ECMWF results is the best fitting
  equation for this layer. Each layer could have the individual optimal polynomial function in
  general.

Fourth, after finding the optimal polynomial function of each layer in different heights, using the
latitude, longitude and altitude information into the function could obtain the three-dimensional water vapor
distribution of any position in the tomographic region. The three-dimensional water vapor field in the
tomographic zone can be described by broadcasting the estimated coefficients of the layered optimal
polynomial functions.
*2.3. The optimal polynomial selection based on adaptive training*
Since the polynomial form can better reflect the continuity of water vapor and has the advantage of
high-efficiency computing as well as easy expression, this paper chooses the polynomial form as the layered
fitting function. The selection process of the layered optimal polynomial function based on adaptive training
is as follows:
First, construct a polynomial equations training library, which contains a wide variety of polynomial
function forms of the latitude and longitude as independent variables while the water vapor density in the
voxels as the dependent variable. After many experiments, the maximum power of the latitude and
longitude found as 5 is sufficient to describe the water vapor changes. Therefore, the maximum power of
the fitting function part is adopted as 5 in the training library.
Second, according to the water vapor density observations from the voxels penetrated by the GNSS
signals at each level, the form of the candidate polynomial function of each layer is automatically
determined from the polynomial function training library to ensure that the number of observations at all
levels is always greater than the number of estimated coefficients of the candidate polynomials.
Third, calculate the water vapor variation index (WVVI) of each layer in both east-west and north-
south directions using the traditional tropospheric tomography results as shown in Eq. (10).
$$WVVI = \frac{\overline{\nabla wv_{EW}}}{\overline{\nabla wv_{NS}}} \tag{10}$$

where $wv_{EW}$ and $wv_{NS}$ are the water vapor density in east-west and north-south direction, separately.

The WVVI, a changing rate indicator of the water vapor density in a given direction, is obtained by

calculating the overall average change rate of the water vapor density in a given direction within each
adjacent voxel. According to the water vapor variation index of each layer in the east-west and north-south
direction, it can be determined whether the water vapor exists mainly in the east-west distribution or the
north-south distribution. As an aid, WVVI can choose the main body of the alternative polynomial function
with higher order term of the longitude or latitude for the subsequent accuracy comparison in order to
efficiently and quickly find the layered optimal polynomial function. If the water vapor density of a layer
indicates a horizontal gradient of east-west distribution, the polynomial function with higher-order term of
the longitude should be given the priority. It suggests that when the water vapor shows an east-west gradient
distribution there is a better correlation between the longitude and the water vapor variation, furthermore
the high-order term in longitude can better reflect the nuanced water vapor variation. A simple example of
the polynomial function with a higher-order term in longitude is shown in Eq. (11):
$$V_d = a_0 + a_1 B + a_2 L + a_3 BL + a_4 L^2 + a_5 BL^2 + a_6 L^3 \qquad (11)$$

Otherwise, when the water vapor density of a layer indicates a horizontal gradient of north-south

distribution, the polynomial function with higher-order term of the latitude should be given the priority. A
simple example is shown in Eq. (12):
$$V_d = a_0 + a_1 B + a_2 L + a_3 BL + a_4 B^2 + a_5 B^2 L + a_6 B^3 \qquad (12)$$

While the distribution regularities of the water vapor density gradient are not clear or obvious, the

polynomial function with the same order of the latitude and longitude can be considered as the example
shown in Eq. (13):
$$V_d = a_0 + a_1 B + a_2 L + a_3 BL + a_4 B^2 + a_5 L^2 \qquad (13)$$

Fourth, the candidate polynomials of all levels screened by the WVVI gradient auxiliary information

are used as the next comparative polynomials, and the required estimated coefficients of the comparative
polynomial are solved according to the principle of least squares through Eq. (14) and automatically
recorded into the coefficients data set. $M$ is the matrix of the longitude and latitude, and the vector $x$
comprises the unknown coefficients of the comparative polynomial functions as shown in Eq. (15).
$$V_d = Mx \qquad (14)$$

$$x = \begin{bmatrix} a_0 \\ a_1 \\ \vdots \\ a_n \end{bmatrix} \qquad (15)$$

Fifth, through the comparative polynomials with the estimated coefficients in each layer, the whole-

voxel water vapor fitting of each layer is automatically fit with the information of the latitude and longitude.
In order to obtain the RMSE, the fitting result would be compared with the ECMWF water vapor density
of each layer in this period. The results are then saved to the accuracy data sets of each layer. The
comparative polynomials with the estimated coefficients are constantly selected to train the fitting of the
layered water vapor density and then compared with the water vapor density of ECMWF at each layer.
Thus, large accuracy data sets of RMSE can be obtained, where the smallest RMSE value of the
comparative polynomial form can be chosen, and then the optimal polynomial of each layer could come
into being. It is noteworthy that the optimal polynomial of each layer might be different. With the layered
optimal polynomial, the three-dimensional water vapor density in the tomographic region can be expressed
conveniently and continuously by transmitting the estimated coefficients information.

## 259     3. Experiment

*3.1. Experimental description and data-processing strategy*
To study whether the accuracy and stability of the results of the improved tropospheric tomography
model are better than the traditional pixel-based tropospheric tomography model, the following experiment
is designed.
Tomographic data is obtained from the SatRef Network for Hong Kong from 25 March 2014 to 25
April 2014. Two epochs are taken each day (0:00 and 12:00 UTC). The corresponding meteorological data
is also used to calculate the PWV. The tomographic area ranges between latitude 22.24°N to 22.54°N and
longitude 113.87°E to 114.29°E. Taking the mean sea level as the height of the base level, the vertical
resolution is 0.8 km, and total grid number is $5 \times 7 \times 13$. In the selected area, a total of 11 GNSS stations
and 1 radiosonde station (located at King's Park, Hong Kong) are selected, and the ECMWF grid data are
extracted twice daily at 00:00 and 12:00 UTC from 25 March 2014 to 25 April 2014 (grid resolution of
$0.125 \times 0.125$). See Fig. 1 for details.

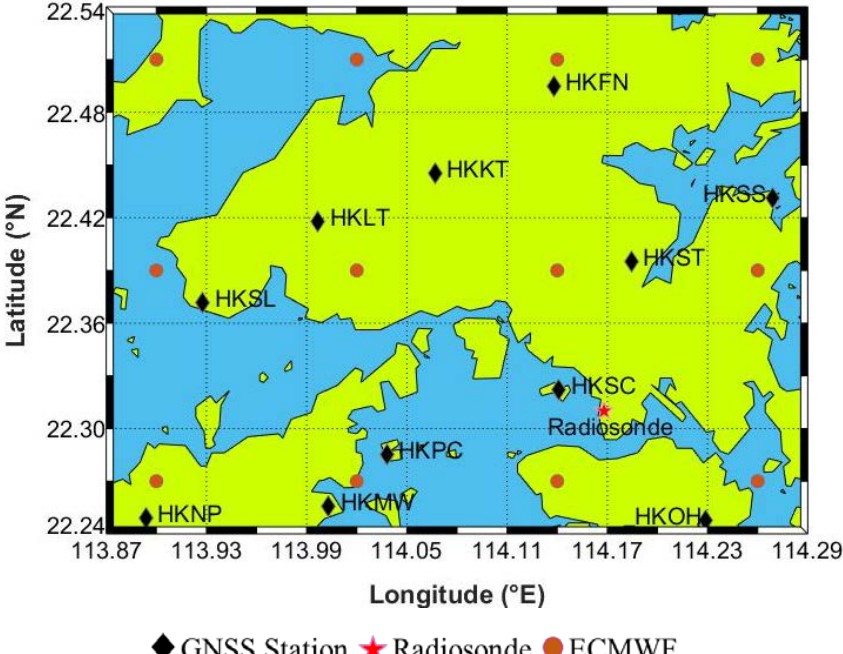



**Figure 1.** The GNSS stations (11 black rhombuses) and the radiosonde station (1 red star) and the ECMWF comparative
points (12 ochre circles) in Hong Kong. The grid lines display tomography grids.
According to the official website of the Hong Kong Observatory
(http://www.weather.gov.hk/contentc.htm) for the weather review, Hong Kong had a total of 15 days of
rainy weather from 25 March 2014 to 25 April 2014, as shown in Table 1.

**Table 1.** Rainfall information for March and April 2014.

| Date | Rainfall situation |
|------|--------------------|
| 3.29 | Thunderstorms turn to heavy rain |
| 3.30 | Thunderstorms turn to heavy rain |
| 3.31 | Thunderstorms turn to heavy rain |
| 4.1 | Showers accompanied by wind, thunderstorms |
| 4.2 | Showers, reports of hail in some areas |
| 4.3 | Showers, some parts of the rain are quite large |
| 4.6 | Cloudy showers, low temperature |
| 4.7 | Heavy showers, low temperature |
| 4.8 | Showers, low temperature |
| 4.14 | Showers |
| 4.21 | Cloudy turns to the showers |
| 4.22 | Showers and foggy |
| 4.23 | Showers turn to the rain |
| 4.24 | Showers turn to the cloudy |
| 4.25 | Cloudy turns to the rain |

In this paper, GAMIT (v10.50) (Herring et al., 2010) software was used for processing the GPS
observations based on the double-differenced model at a sampling interval of 30 s, and the global mapping
function was adopted. The zenith total delay (ZTD) and wet horizontal gradient intervals were estimated at
0.5 h and 2 h, respectively. Based on the surface pressure obtained from observed meteorological
parameters, the ZHD could be obtained by the Saastamoinen model, and ZWD was isolated from ZHD.
Via GMF projection, the SWD could be obtained by transforming the observed SWV.
*3.2. Experimental introduction and program comparison*
The RMSE and bias of the improved tomography model residuals were calculated by subtracting the
ECMWF water vapor density from the water vapor density of the improved pixel-based water vapor
tomography model (hereinafter referred to as improved tomography model). In a similar way, the RMSE
and bias of the traditional tomography model residuals can also be obtained from the difference between
the ECMWF water vapor density and the three-dimensional water vapor density obtained by the traditional
pixel-based tropospheric tomography model (hereinafter referred to as the traditional tomography model).
In the period of data processing, the situation can be compared on a case-by-case basis to
comprehensively evaluate the accuracy of the improved tomography model from various views. In this
paper, 6 scenarios are investigated, comprising the spatial distribution scenario, the everyday distribution
scenario, the rainy scenario and the non-rainy scenario. Moreover the residuals of the water vapor density
in voxels with and without penetrating GNSS rays are inspected. The definitions of 6 scenarios
abovementioned are as follows:
The spatial distribution scenario is investigated by obtaining the RMSE and bias of the residuals from
all ECMWF comparative points at all time intervals as well as the layered tomography accuracy.
The everyday distribution scenario is found by obtaining the RMSE and bias of the residuals from all
ECMWF comparative points in two epochs each day, and the overall accuracy of 32 days between 25 March
2014 and 25 April 2014 was calculated.
The rainy scenario is based on the distribution of 15 days of rainy days between 25 March and 25
April, 2014, as referred to in Table 1, in which the RMSE and bias of the residuals are obtained from all
ECMWF comparative points in all the epochs in rainy days for the further accuracy analysis. Similarly, the
non-rainy scenario is found with the accuracy analysis of the non-rainy days.

The scenario of residuals of the water vapor density in voxels without GNSS rays penetration is found

by obtaining the RMSE and bias of the residuals from ECMWF comparative points without rays passing
through in all the epochs each day. Conversely, the scenario with GNSS rays penetration is found by
obtaining the RMSE and bias of the residuals from ECMWF comparative points with rays passing through
in all the epochs each day.

According to the above classifications, the accuracy of the improved tomography model residuals and

the traditional tomography model residuals were calculated, and the accuracy of the new model was
compared with the traditional model to determine which one is better. Furthermore, the accuracy
comparison of the water vapor density derived from two models and radiosonde data was designed to show
if the improved model would be more efficient than the traditional one.
**4. Interpretation of 6 scenario results**
*4.1. Accuracy information of the spatial distribution scenario*

To verify whether the accuracy of the improved tomography model is better than that of the traditional

tomography model, the layered RMSE and bias of the residuals from all ECMWF comparative points at all
time intervals between the tomography (using both the optimal polynomial function of each layer and the
traditional way) and the ECMWF results are obtained and shown in Table 2, and the calculation of RMSE
improvement percentage involved in the following tables is shown in Eq. (16).
$$\Delta RMSE\% = \left( RMSE_{trad} - RMSE_{impr} \right) / RMSE_{trad} \cdot 100\% \quad (16)$$
where $RMSE_{impr}$ is the RMSE value of the residuals calculated from the improved tomography model,
and $RMSE_{trad}$ is the RMSE value of the residuals obtained from the traditional tomography model.

Table 2 shows that RMSE and bias values obtained by the improved tomography model are smaller

than those of the traditional tomography model, and the RMSE improvement percentage is positive, which
indicates that the improved tomography model has a higher accuracy than the traditional tomography model
overall. Moreover, the RMSE improvement percentage is appreciable in the upper region because the value
of the water vapor density in high altitude is very small (see Fig. 2 for details), even the small changes in
the upper region could result in a large percentage change. In addition, the bias and RMSE in the bottom
from Table 2 are not as good as those of the other higher layers, regardless of which model is used. These
results could be mainly ascribed to a certain system deviation between the comparison data of ECMWF
and the GNSS tomographic data. Besides, the observations and the number of redundant observations are
insufficient due to less voxels with GNSS rays penetration in the bottom, resulting in the low accuracy.
What's more, Figure 2 shows that the water vapor content in the bottom region is too abundant and
changeable to be generally described accurately. These above reasons lead to large bias and RMSE values
in the bottom tropospheric area.
**Table 2.** Statistics of two models' tomography accuracy with respect to ECMWF data in the spatial distribution scenario
for the experimental period (Unit: g/m³).

| Layer | bias | | RMSE | | RMSE Improvement Percentage |
|---|---|---|---|---|---|
| | Traditional model | Improved model | Traditional model | Improved model | |
| 1 | -7.81 | -7.65 | 8.17 | 8.00 | 2.06% |
| 2 | -3.52 | -3.42 | 3.95 | 3.83 | 3.14% |
| 3 | -0.90 | -0.80 | 1.66 | 1.60 | 4.05% |
| 4 | 0.72 | 0.61 | 1.39 | 1.36 | 2.00% |
| 5 | 1.62 | 1.58 | 1.87 | 1.83 | 2.28% |
| 6 | 1.95 | 1.77 | 2.10 | 2.09 | 0.39% |
| 7 | 1.98 | 1.90 | 2.25 | 2.20 | 2.07% |
| 8 | 1.76 | 1.68 | 2.15 | 2.10 | 2.32% |
| 9 | 1.62 | 1.60 | 2.06 | 2.04 | 1.10% |
| 10 | 1.34 | 1.11 | 1.85 | 1.47 | 20.68% |
| 11 | 1.04 | 0.87 | 1.60 | 1.25 | 21.75% |
| 12 | 0.74 | 0.61 | 1.26 | 0.96 | 23.67% |
| 13 | 0.44 | 0.38 | 0.71 | 0.58 | 18.36% |

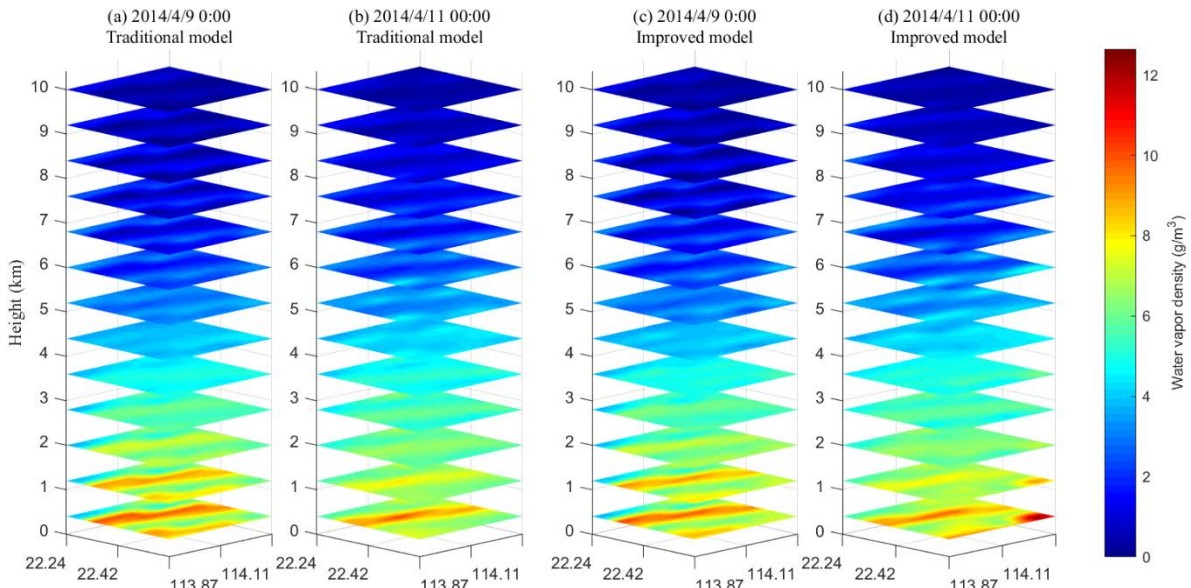

**Figure 2.** The layered maps of the water vapor density from **(a) (b)** the traditional model and **(c) (d)** the improved
model at specific epochs, **(a) (c)** 0:00 UTC 9 April 2014 and **(b) (d)** 0:00 UTC 11 April 2014.
*4.2. The accuracy information of the everyday distribution scenario*

To determine whether the accuracy of the improved tomography model is better than that of the
traditional tomography model on the everyday time scale, the RMSE improvement percentage is obtained
from all ECMWF comparative points (a total of 12) at two epochs each day using both the layered optimal
polynomial functions and the traditional method. Figure 3 shows that the percentage of RMSE improvement
per day is practically positive, and the percentage of April 11th can even approach 12%, indicating that the
improvement seems to be appreciable. This improvement shows that the accuracy of the improved
tomography model is mostly superior to that of the traditional tomography model in everyday distribution;

however, on April 7, April 9 and April 15, the RMSE improvement percentage is negative. This might be due to the heavy showers bringing rapid water vapor changes from April 7 to April 8 and on April 14, which is difficult to fit the polynomial function well with the unstable water vapor. However, since negative percentages do not exceed -1%, the accuracy of these four days calculated by the improved tomography model could be considered basically equivalent to that of the traditional tomography model.

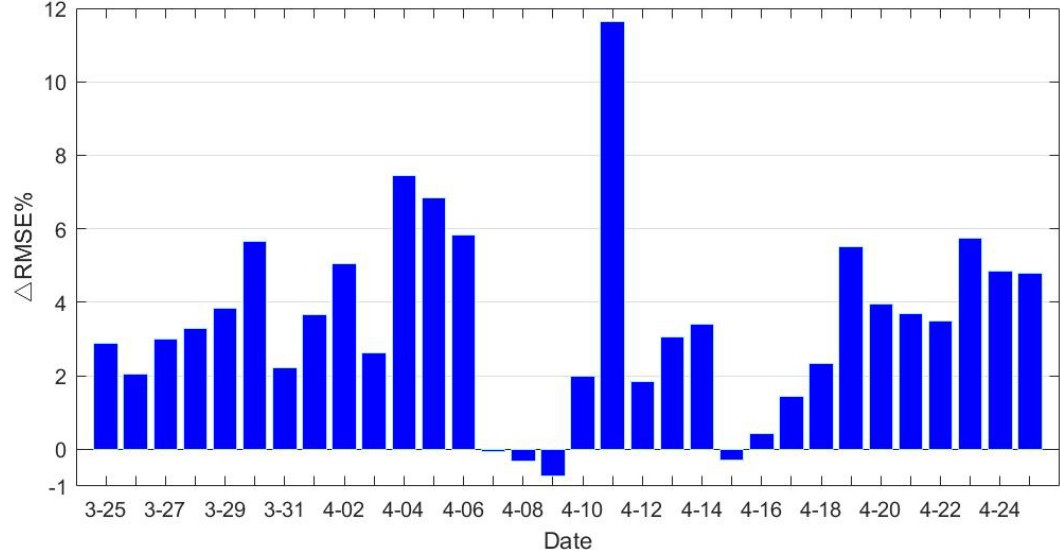

**Figure 3.** Everyday distribution statistics of daily RMSE improvement percentage between 25 March and 25 April, 2014.

In addition, the overall RMSE and bias of the residuals are obtained from the ECMWF comparative points (a total of 12) in two epochs under the entire everyday distribution scenario. The statistical results are shown in Table 3 below.

**Table 3.** Statistics of two models' tomography accuracy with respect to ECMWF data in the everyday distribution scenario for the experimental period (Unit: $g/m^3$).

| Statistics type | Traditional model | Improved model | RMSE improvement percentage |
|---|---|---|---|
| RMSE | 2.97 | 2.87 | 3.44% |
| bias | 0.07 | 0.02 | |

Table 3 shows that the RMSE obtained by the improved model is smaller by 3.44% compared to the traditional one. The bias of the improved model more closes to zero, indicating that the improved tomography model has better stability and less systematic deviation from the comparative data. The better accuracy of the improved model compared to the traditional one illustrates the edge of the improved model.

*4.3. The accuracy information of rainy and non-rainy scenarios*

To further analyze the reliability of the improved tomography model compared with the traditional tomography model in different weather conditions, according to the distribution of rainy days in Table 1, all the rainy days data and non-rainy days data are used separately for tomography to obtain the RMSE and bias of the residuals under corresponding weather conditions. The number of matching points is still 12 (see Fig. 1). The overall statistical results are shown in Table 4.

**Table 4.** Statistics of two models' tomography accuracy with respect to ECMWF data in the rainy scenario and the non-
rainy scenario for the experimental period (Unit: $g/m^3$).

| (a) The overall rainy scenario statistics | | | |
|---|---|---|---|
| Statistics type | Traditional model | Improved model | RMSE improvement percentage |
| RMSE | 3.05 | 2.94 | 3.68% |
| bias | 0.05 | -0.01 | |
| (b) The overall non-rainy scenario statistics | | | |
| Statistics type | Traditional model | Improved model | RMSE improvement percentage |
| RMSE | 2.89 | 2.80 | 3.21% |
| bias | 0.10 | 0.04 | |

Table 4 (a) shows that the RMSE and bias of the residuals calculated by the improved tomography

model are better than those of the traditional tomography model using rainy days' data. The RMSE of the
improved tomography model is 3.68% higher than that of the traditional model, indicating the accuracy of
the new model is superior. The improved model bias closes more to zero than that of the traditional one,
which means the new model has an increase in stability and a reduction in the system error. Using non-
rainy days' data, the RMSE and bias of the residuals calculated by the improved tomography model are
also better than those of the traditional tomography model, see Table 4 (b). The RMSE improvement
percentage is 3.21%, also indicating there is an improvement in the accuracy of the new model. Besides,
the improved model bias is more close to zero, making the system error weakened and the stability enhanced.
According to the RMSE improvement percentage under the rainy and non-rainy scenarios, the RMSE
improvement percentage of rainy days is better than that of non-rainy days. This finding shows that the
improved tomography model is more suitable for obtaining the tomographic results when severe water
vapor changes occur.
*4.4. The accuracy information of voxels with and without GNSS rays penetrating scenarios*

In the traditional pixel-based water vapor tomography model, the water vapor density in the voxels

without GNSS rays passing through depends on the accuracy of the water vapor density in the adjacent
voxels with GNSS rays penetration. However, the improved tomography model uses the layered optimal
polynomial function for overall fitting to obtain the water vapor density in voxels without penetrating GNSS
rays. To determine whether the layered optimal polynomial function of the improved method contributes
better to the accuracy of the water vapor density, the scenarios of voxels with and without GNSS rays
penetration as described in section 3.2 were designed. After obtaining the RMSE and bias of the residuals
using the improved and traditional tomography models separately under designed scenarios, the overall
accuracy information of voxels with and without GNSS rays penetrating shows in Table 5.

 **Table 5.** Statistics of two models' tomography accuracy with respect to ECMWF data in the voxels with and without

 penetrating GNSS rays for the experimental period (Unit: g/m$^3$).

| (a) The overall scenario statistics of voxels without rays penetrating | | | |
|---|---|---|---|
| Statistics type | Traditional model | Improved model | RMSE Improvement Percentage |
| RMSE | 3.40 | 3.20 | 5.88% |
| Bias | 1.59 | 1.51 | |
| (b) The overall scenario statistics of voxels with rays penetrating | | | |
| Statistics type | Traditional model | Improved model | RMSE Improvement Percentage |
| RMSE | 3.27 | 3.24 | 1.00% |
| bias | 1.70 | 1.65 | |

Table 5 (a) shows that the RMSE and bias of the residuals calculated by the improved tomography model are better than those of the traditional tomography model in the scenario of voxels without GNSS rays penetrating. Moreover the RMSE of improved tomography model is 5.88% better than that of the traditional tomography model, and the bias decreased from 1.59 to 1.51 g m$^{-3}$. To a certain extent, this finding shows that the improved tomography model is more advantageous for obtaining the water vapor density from the voxels without GNSS rays penetrating, which is consistent with the initial hypothesis: the traditional tomography model uses empirical constraint equations in section 2.1.2, Eq. (7), which is unable to well represent the distribution of the water vapor density from voxels without GNSS rays penetrating in the actual situation. However, the new proposed model uses the relatively exact water vapor density from voxels with GNSS rays penetrating as the observation values to further fit the water vapor density in voxels without GNSS rays penetrating. Therefore, the improved tomography model can better reflect the actual layered situation of continuous water vapor changes, and the accuracy is naturally better. What's more, the RMSE and bias obtained by the improved tomography model are also superior to those of the traditional tomography models using the classified data of voxels with GNSS rays penetrating, see Table 5 (b). The RMSE calculated by the new model is 1% higher than that of the traditional model, and the bias reduced from 1.7 to 1.65 g m$^{-3}$. In summary, whether it is calculated separately from data of voxels with or without GNSS rays penetrating, the results of the improved tomography model are superior to those of the traditional tomography model to a certain extent, which could prove the advanced nature and reliability of the improved tomography model.

In order to double-check if the improved tomography model in the scenario of voxels without GNSS rays passing through shows a better result in the vertical distribution of the three-dimensional water vapor density, the water vapor density profiles for different altitudes at individual times are given in Fig. 4. Two times (0:00 UTC 11 April 2014 and 12:00 UTC 11 April 2014) are chosen for they correspond to the maximum percentage of RMSE improvement during the experiment period of 32 days. Figure 4 shows that in the scenario of voxels without GNSS rays penetration, the water vapor profile of the improved tomography model better matches that of ECMWF data than the traditional tomography model at both times, especially in the bottom layers, which again implies that the water vapor density derived from the improved model is superior to that of the traditional one in the scenario of voxels without penetrating GNSS

rays.

432

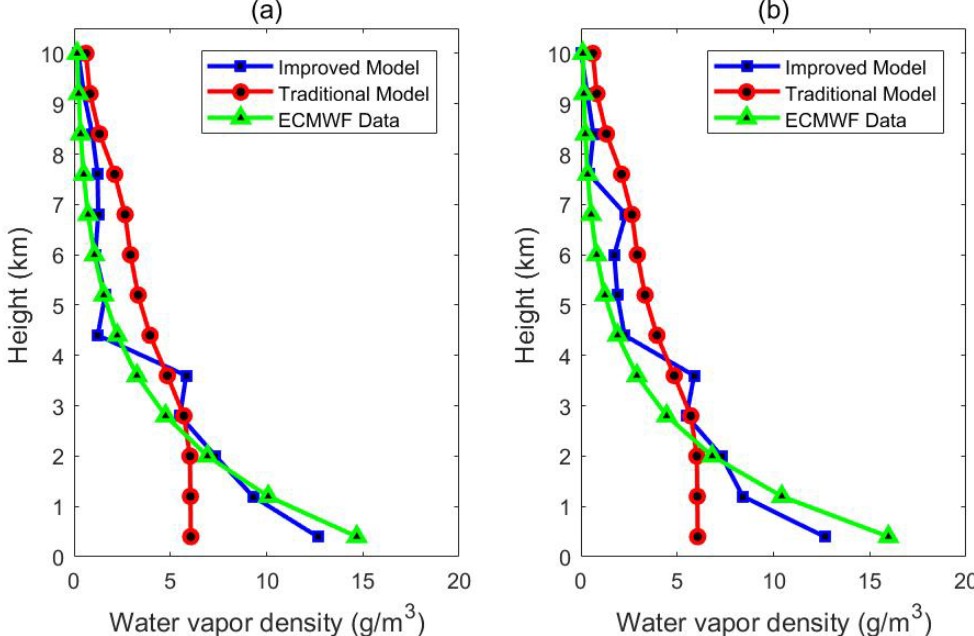

**Figure 4.** Water vapor profiles derived from ECMWF and two models in the scenario of voxels without penetrating GNSS rays, **(a)** and **(b)** are periods of 0:00 UTC 11 April 2014 and 12:00 UTC 11 April 2014, respectively.

      Furthermore, to compare directly the vertical accuracy of the water vapor density derived from different altitudes in the scenario of voxels without penetrating GNSS rays, the tomographic results (25 March 2014 to 25 April 2014) from two different tomography models are analyzed. Figure 5 shows the percentage of RMSE improvement and the relative error of the water vapor density changing with altitudes. The percentage of RMSE improvement in Fig. 5 is defined as the same as Eq. (16), and the relative error is defined by using the Eq. (17).

$$RE = \frac{\rho - \rho_{ECMWF}}{\rho_{ECMWF}} \tag{17}$$

      where *RE* is the relative error, $\rho$ represents the water vapor density derived from the traditional or improved tomography model, and $\rho_{ECMWF}$ is the water vapor density derived from ECMWF grid data.

      It can be observed in Fig. 5 that in the scenario of voxels without GNSS rays penetration the percentage of RMSE improvement is positive in lower layers while negative in some middle and upper layers, which could prove that the proposed model improves the accuracy of tomography results in most layers when there are seldom voxels with GNSS rays penetrating especially in the bottom layers. Due to the lack of GNSS observation data, the bottom accuracy of tomographic results is generally low. In addition, Figure 5 shows in the scenario of voxels without GNSS rays penetration, the relative error begins to decrease with the altitude and then increases above 3 km. When the altitude is higher, the relative error becomes larger because of the small water vapor values of the upper layers, a very tiny difference could cause a large relative error between the models and the ECMWF data.

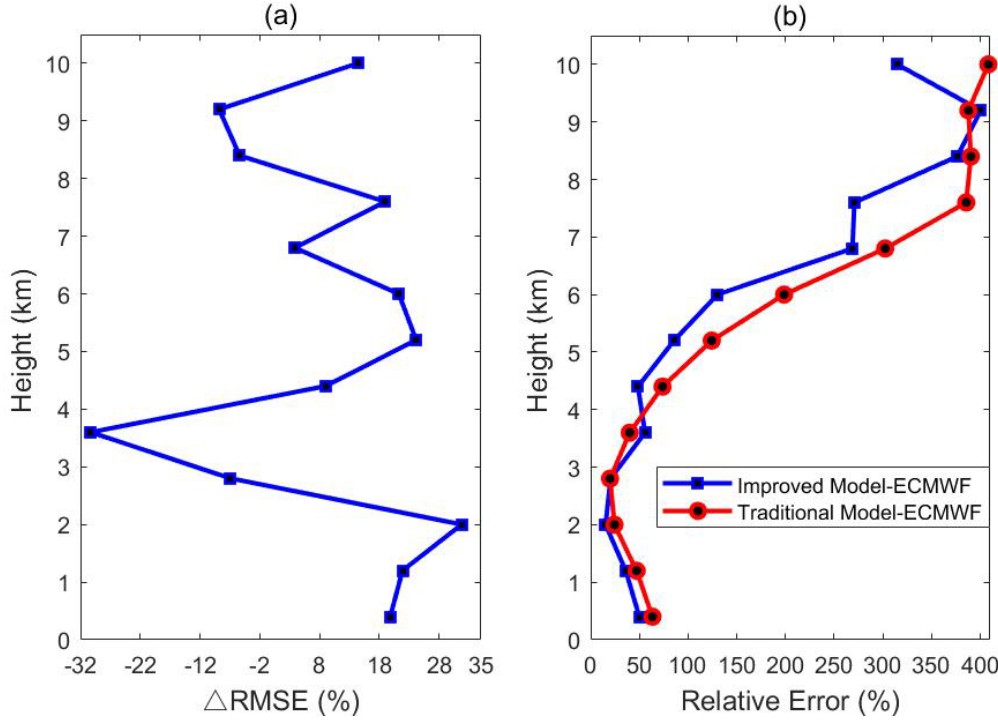

**Figure 5.** In the scenario of voxels without GNSS rays penetration **(a)** the percentage of RMSE improvement and **(b)** the relative error change with height (the blue curve and red curve are derived from the differences between the profiles of the improved tomography model, the traditional tomography model and ECMWF grid data, separately for 64 epochs from 25 March 2014 to 25 April 2014).

## 5. Water vapor comparison with radiosonde data

As radiosonde data can provide fairly accurate vertical profiles of tropospheric water vapor (Niell et al., 2001), in this paper, the water vapor profiles derived from radiosonde data, as a reference, are used to validate the tomographic results from two models for showing if the improved model would be more efficient than the traditional one. In Hong Kong, there is one radiosonde station located at King's Park (shown in Fig. 1) where radiosonde balloons are launched twice daily at 0:00 and 12:00 UTC, respectively. The water vapor profiles derived from the improved model and the traditional model for the location of the radiosonde station are compared with that from radiosonde data at 00:00 and 12:00 UTC daily for the experimental period of 32 days. The overall statistical results are shown in Table 6. The RMSE and the bias of the improved model are 2.24 and -0.34 g m$^{-3}$, respectively, and the values using the traditional model are 2.13 and -0.46 g m$^{-3}$, respectively, which indicates that the RMSE of the improved model is not as good as the traditional model while the bias of the improved model is a little better than that of the traditional one. The reason for poor accuracy of the improved model could be due to systematic differences between the training source ECMWF data and the radiosonde data as the water vapor density of the improved model is obtained by the optimal polynomial selection based on adaptive training with ECMWF data. Besides, shown in Fig.1, the location of the radiosonde station is close to one GNSS station (HKSC), leading to the voxels for the location of the radiosonde station having GNSS rays penetration. Since the improved model has advantages of obtaining water vapor density just from voxels without GNSS rays penetration, this situation cannot show the superiority of the improved model.

**Table 6.** Statistics of two models' tomography accuracy with respect to radiosonde data for the experimental period
(Unit: g/m³).

| Statistics type | Traditional model | Improved model |
|---|---|---|
| RMSE | 2.13 | 2.24 |
| bias | -0.46 | -0.34 |

In addition, water vapor profiles obtained by two models and radiosonde data are compared for the
specific two epochs at 0:00 UTC 25 March 2014 and 0:00 UTC 7 April 2014, shown in Fig. 6. Those two
times are selected because they correspond to the non-rainy day and heavy rainfall day, which could be
more comprehensive and representative for the comparison results of water vapor profiles. It can be seen
from Fig. 6 that no matter in the non-rainy day or the rainy day, both the improved model and the traditional
model can hardly match the radiosonde data at most altitudes, especially at the lower layers, showing again
this radiosonde data comparison experiment design cannot reflect the superiority of the improved model.
However, Figure 6 also shows the water vapor profiles of the improved model almost match that of the
traditional model, whether it is non-rainy or rainy, indicating that though both under the situation of poor
water vapor profile matching results the improved model still has the advantage of the convenient and
efficient expression.

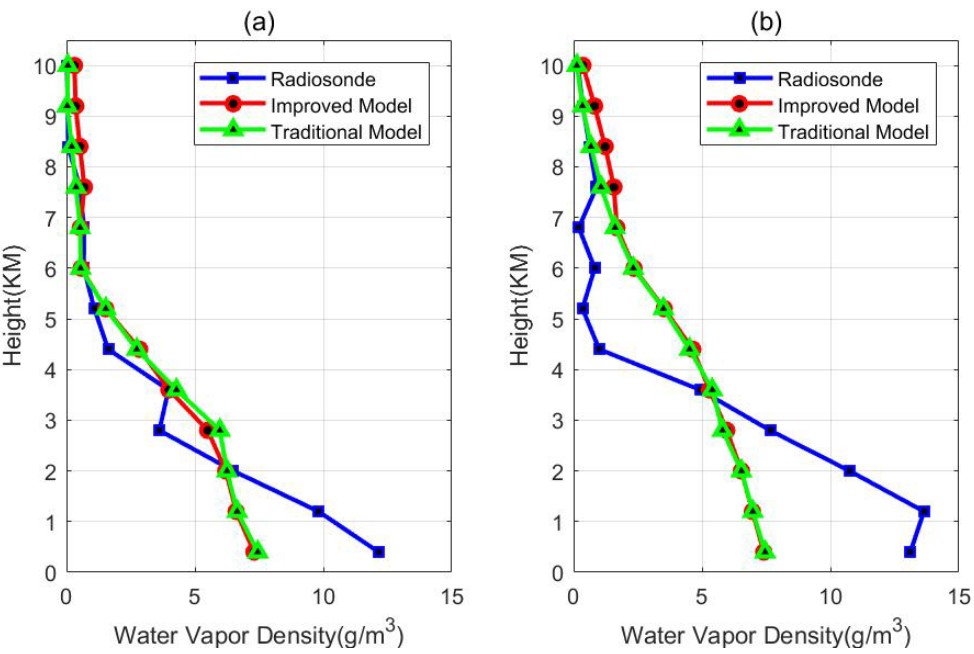

**Figure 6.** Water vapor profile comparison derived from different tomographic methods and radiosonde, **(a)** a non-rainy
day at 0:00 UTC 25 March 2014, **(b)** a rainy day at 0:00 UTC 7 April 2014.
**6. Conclusion**
In this paper, an improved pixel-based water vapor tomography model has been proposed, which is
much more concise and convenient in expression than the traditional one. Only the optimal polynomial
coefficients of each layer are required to describe the three-dimensional water vapor distribution in the
tomographic region. By using the SatRef GNSS network observation data in Hong Kong between 25 March
and 25 April, 2014, the RMSE and bias have been assessed in 6 scenarios. The scenarios include the spatial
distribution scenario and the everyday distribution scenario, the rainy scenario and the non-rainy scenario,
and the voxels with and without GNSS rays penetrating scenarios. The results demonstrate that in either
case, the RMSE and bias of the improved tomography model are better than that of the traditional
tomography model. Among these scenarios, when there are voxels without GNSS rays penetrating, the
RMSE improvement percentage can be significantly increased up to 5.88%, which shows that the improved
tomography model is more advantageous for obtaining the water vapor density from voxels without GNSS
rays penetration. Using radiosonde data for evaluation, it is proved that with the almost similar accuracy
the improved model is more efficient in expression than the traditional one. However, some shortcomings
remain in the improved GNSS tropospheric tomography model. For example, when constructing the
optimal polynomial of each layer, the polynomial is not only limited by the water vapor density quality in
voxels with GNSS rays passing through calculated by the traditional pixel-based tomography model, but it
is also limited by the size of the tomographic area and the situation of dividing voxels. In the future, the
function-based water vapor tomography model should be further studied, which is free from the above
limitations. It is expected that the function-based tropospheric tomography model will be more conveniently
used when the expression parameters of the function part could be obtained directly from SWVs.
**Acknowledgments:** The authors would like to thank ECMWF for providing access to the layered meteorological data.
The Lands Department of HKSAR is also acknowledge for providing GPS data from the Hong Kong Satellite
Positioning Reference Station Network (SatRef) and corresponding meteorological data.
**Conflicts of Interest:** The authors declare that they have no conflict of interest.

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
