# Peer review of "An improved pixel-based water vapor tomography model"

_Annales Geophysicae, 2018_

## Referee Comment (RC1) · Anonymous Referee #1 · 10 Jul 2018

Review angeo-2018-34

The authors have tried to estimate the amount of water vapor at any desired point in the tomography area. For this purpose, they have used the results of the tomography technique. In other words, a kind of interpolation is used to calculate the water vapor density at the points among the centers of tomographic voxels. This article is well written but now, after many years of using the troposphere tomography, this technique requires more innovations. Using different methods of interpolation has always been inevitable since the voxel-based tomography process. The main problems are:

a. Function-based tomography is the direct calculation of water vapor density using signal delays at arbitrary points and it is independent of the voxel-based method. In this paper, the voxel-based tomography has been used to model the water vapor density

at each voxel and water vapor density has not been calculated directly at arbitrary points. Therefore, the use of the phrases "function based model" or "function based tomography" is not correct.

b. Why did not authors use the function based method directly and without voxel-based tomography? In other words, why the slant tropospheric delay (SWD) of the signals is not considered as a function of the geographical location?

c. In this paper, the polynomial function is used for interpolation. This interpolation method causes fluctuations between interpolation points. Due to the small size of the study area, these fluctuations increase the error of interpolation results between interpolation points. In these cases, other interpolation functions or method with less variation between interpolation points could be used.

d. In order to show the efficiency of the proposed method, it is better to give the map of water vapor density from voxel-based tomography and from the paper method.

e. Due to the presence of the radiosonde station in the study area, it is necessary to compare the results of voxel-based tomography and result of the paper method with radiosonde observations to show that the proposed method in this paper is more efficient than voxel-based tomography.

f. In this paper, many self-citations have been used. Also, in the introduction section authors did not refer to the new and valid articles, which used new techniques in different steps of tomography such as choosing the best dimensions for Voxels, Applying 3d ray tracing, using AIRS measurements, for example: "HajiAghajany, S., Amerian, Y. (2017). Three-dimensional ray tracing technique for tropospheric water vapor tomography using GPS measurements. Journal of Atmospheric and Solar-Terrestrial Physics, Volume 164, 2017, Pages 81-88. "

g. The first and appropriate references for virtual stations topics are the follows: * Vollath U, Buecherl A, Landau H, Pagels C, Wager B (2000) Multibase RTK positioning

using virtual reference stations.In: Paper presented at the Proceedings 13th International Technical Meeting of the Satellite Division of the US Institute of Navigation, ION GPS-2000, Salt Lake City, September, 19–22. * Marel H-v-d (1998) Virtual GPS reference stations in the Netherlands. In: Paper presented at the Proc 11th International Technical Meeting of the Satellite Division of the US Institute of Navigation, ION GPS-98, Nashville, TN, September 15–18.

---

## Referee Comment (RC2) · Anonymous Referee #2 · 7 Aug 2018

angeo-2018-34

Function-based tomography is one of the most important issues in the field of GNSS. This paper addresses this issue. But the method described in this paper is still dependent on high precision voxel-based tomography. This paper can be published after major revision. Because the authors have written and discussed the subject well.

1- Why did the authors use the polynomial function for this purpose?

2- Due to the important role of the voxel-based tomography technique in this method, the use of the term "function-based tomography" may lead to misinterpretation. For this reason, it is recommended that the title of the paper be changed.

3- Why the results have not been compared with the radiosonde observations?

4- Due to the importance of the issue, the authors should draw up and compare the results obtained from the two methods.

5- References need to be revised. For example, (adavi and mashhadi-hossainali, 2015) is not a valid and appropriate reference. Also, there are no valid and new articles on tomography and its accuracy in the list of resources (except self citations).

Please also note the supplement to this comment:
https://www.ann-geophys-discuss.net/angeo-2018-34/angeo-2018-34-RC2-supplement.pdf

---

## Author Comment (AC1) · 12 Sep 2018

We thank the referee #1 for the insightful comments and constructive suggestions. We have addressed all their comments in the revised manuscript. Below are our responses to the referee's critical comments (*Italics*). The page and line numbers in our responses refer to those in the revised manuscript.

*a. Function-based tomography is the direct calculation of water vapor density using signal delays at arbitrary points and it is independent of the voxel-based method. In this paper, the voxel-based tomography has been used to model the water vapor density at each voxel and water vapor density has not been calculated directly at arbitrary points. Therefore, the use of the phrases "function based model" or "function based tomography" is not correct.*

Authors: Thanks for the reviewer's reminding, the phrases "function-based model" and "function-based tomography" in our new proposed model has been revised and deleted. The title has changed into "An improved pixel-based water vapor tomography model". Only the future work of the function-based tomography model in the conclusion part has been preserved in this paper. These phrases has been revised throughout the manuscript.

*b. Why did not authors use the function based method directly and without voxel-based tomography? In other words, why the slant tropospheric delay (SWD) of the signals is not considered as a function of the geographical location?*

Authors: Thanks for the reviewer's question, to our knowledge, since the tropospheric tomography has been proposed, there are few pure function-based water vapor tomography models in the previous research. It's too challenging to obtain the water vapor density directly from the slant tropospheric delay of the signals so far. In this paper we just try to find out if the function part could be used in the tropospheric tomography model. As now the results of this paper turn out to be good, in the near future we will dedicate ourselves to building the pure function-based water vapor tomography model, which would consider the SWD of signals as a function of the geographical location.

*c. In this paper, the polynomial function is used for interpolation. This interpolation method causes fluctuations between interpolation points. Due to the small size of the study area, these fluctuations increase the error of interpolation results between interpolation points. In these cases, other interpolation functions or method with less variation between interpolation points could be used.*

Authors: Thanks for the reviewer's reminding, however, as for the function part of the interpolation method, we did try some other interpolation methods using the 1stOpt (First Optimization) software in previous preparations for the experiments. The results of other interpolation methods were similar with or a little worse than that of the polynomial function. Since the Hong Kong Satellite Positioning Reference Station Network (SatRef) is a flat GNSS network (Zhang et al., 2017), there is no large difference in tomography results between the polynomial function and other interpolation methods while the polynomial function has the easier and more convenient expression. So in this paper we choose the polynomial function for the interpolation.

Reference: Zhang Bao, Fan Qingbiao*, Yao Yibin, Xu Caijun and Li Xingxing. An Improved

Tomography Approach Based on Adaptive Smoothing and Ground Meteorological Observations. Remote Sensing, 2017, 9, 886, DOI:10.3390/rs9090886.

*d. In order to show the efficiency of the proposed method, it is better to give the map of water vapor density from voxel-based tomography and from the paper method.*
Authors: Thanks for the reviewer's reminding, the maps of water vapor density from the traditional tomography model and the proposed tomography model have been presented (Page 10, Line 332, Lines 338-340; Page 11, Figure 2).

*e. Due to the presence of the radiosonde station in the study area, it is necessary to compare the results of voxel-based tomography and result of the paper method with radiosonde observations to show that the proposed method in this paper is more efficient than voxel-based tomography.*
Authors: Thanks for the reviewer's suggestion, the water vapor comparison with radiosonde data section has been added. The comparison results showed that the proposed tomography model was not as good as the traditional tomography model on RMSE and we analyzed the reasons. The main reason could be due to systematic differences between the training source ECMWF data and the radiosonde data as well as the location of the radiosonde station being close to the HKSC GNSS station, leading to the voxels for the location of the radiosonde station having GNSS rays penetration, which is not suitable for the improved tomography model to show its good advantage in the scenario of voxels without GNSS rays penetration. However, the water vapor profiles of the improved model almost match that of the traditional model (Page 17, Figure 6), indicating that the improved model still has the advantage of the convenient and efficient expression (Page 1, Lines 29-30; Page 10, Lines 315-317; Page 16, Line 457 to Page 17, Line 490; Page 18, Lines 503-504).

*f. In this paper, many self-citations have been used. Also, in the introduction section authors did not refer to the new and valid articles, which used new techniques in different steps of tomography such as choosing the best dimensions for Voxels, Applying 3d ray tracing, using AIRS measurements, for example: "HajiAghajany, S., Amerian, Y. (2017). Three-dimensional ray tracing technique for tropospheric water vapor tomography using GPS measurements. Journal of Atmospheric and Solar-Terrestrial Physics, Volume 164, 2017, Pages 81-88. "*
Authors: Thanks for the reviewer's reminding, the introduction section has been rewritten. Some self-citations were deleted and the paper of Haji Aghajany, S. and Amerian, Y. was cited. Besides, we added some new and valid articles for reference in the introduction section (Page 2, Line 54, Lines 67-72).

*g. The first and appropriate references for virtual stations topics are the follows: * Vollath U, Buecherl A, Landau H, Pagels C, Wager B (2000) Multibase RTK positioning using virtual reference stations.In: Paper presented at the Proceedings 13th International Technical Meeting of the Satellite Division of the US Institute of Navigation, ION GPS-2000, Salt Lake City, September, 19–22. * Marel H-v-d (1998) Virtual GPS reference stations in the Netherlands. In: Paper presented at the Proc 11th International Technical Meeting of the Satellite Division of the US Institute of Navigation, ION GPS-98, Nashville, TN, September 15–*

*18.*

Authors: Thanks for the reviewer's reminding, the references for virtual stations topics as you suggested have been rewritten (Page 2, Lines 72-74).

---

## Author Comment (AC2) · 12 Sep 2018

**Responses to Referee #2**

We thank the referee #2 for the insightful comments and constructive suggestions. We have addressed all their comments in the revised manuscript. Below are our responses to the referee's critical comments (*Italics*). The page and line numbers in our responses refer to those in the revised manuscript.

*1- Why did the authors use the polynomial function for this purpose?*

Authors: Thanks for the reviewer's question, in the previous preparations for the experiments, some other plane fitting functions had been tried by the 1stOpt (First Optimization) software and the results were similar with or a little worse than that of the polynomial function. Since the Hong Kong Satellite Positioning Reference Station Network (SatRef) is a flat GNSS network (Zhang et al., 2017), there is no large difference in tomography results between the polynomial function and other interpolation methods while the polynomial function has the easier and more convenient expression. So in this paper we use the polynomial function for this purpose.

Reference: Zhang Bao, Fan Qingbiao*, Yao Yibin, Xu Caijun and Li Xingxing. An Improved Tomography Approach Based on Adaptive Smoothing and Ground Meteorological Observations. Remote Sensing, 2017, 9, 886, DOI:10.3390/rs9090886.

*2- Due to the important role of the voxel-based tomography technique in this method, the use of the term "function-based tomography" may lead to misinterpretation. For this reason, it is recommended that the title of the paper be changed.*

Authors: Thanks for the reviewer's reminding, the term of "function-based tomography" has been changed into "improved tomography" throughout the manuscript. The title of the paper was changed into "An improved pixel-based water vapor tomography model" (Page 1, Lines 1-2).

*3- Why the results have not been compared with the radiosonde observations?*

Authors: Thanks for the reviewer's suggestion, we have added the water vapor comparison with radiosonde data section. The comparison results showed that the proposed tomography model was not as good as the traditional tomography model on RMSE and we analyzed the reasons. The main reasons could be due to systematic differences between the training source ECMWF data and the radiosonde data as well as the location of the radiosonde station being close to the HKSC GNSS station, leading to the voxels for the location of the radiosonde station having GNSS rays penetration, which is not suitable for the improved tomography model to show its good advantage in the scenario of voxels without GNSS rays penetration. However, the water vapor profiles of the improved model almost match that of the traditional model (Page 17, Figure 6), indicating that the improved model still has the advantage of the convenient and efficient expression (Page 1, Lines 29-30; Page 10, Lines 315-317; Page 16, Line 457 to Page 17, Line 490; Page 18, Lines 503-504).

*4- Due to the importance of the issue, the authors should draw up and compare the results obtained from the two methods.*

Authors: Thanks for the reviewer's reminding, the layered maps of the water vapor density

from the two models as you suggested have been compared and presented (Page 10, Line 332, Lines 338-339; Page 11, Figure 2).

*5- References need to be revised. For example, (adavi and mashhadihossainali, 2015) is not a valid and appropriate reference. Also, there are no valid and new articles on tomography and its accuracy in the list of resources (except self citations).*

Authors: Thanks for the reviewer's reminding, the paper of adavi and mashhadihossainali was deleted and changed into other appropriate papers relating to the virtual reference station approach. Besides, some valid and new articles on tomography and its accuracy were added as you suggested (Page 2, Line 54, Lines 67-74).

---

## Referee Report (RR1)

**Review angeo-2018-34**

The paper has been greatly improved. Most comments have been reviewed by authors. This paper can be published after some minor revisions.

a. It is necessary to mention a summary of the authors' responses (a,b,c) in the text.

b. The reference "Adavi and Mashhadi" is not used in the text. Please delete this reference from the list of references.

c. Please consult with a native speaker on the language of the paper to complete the paper's corrections.

---

## Author Response (AR2)

**Responses to Referee #1**

We have tried our best to answer all the questions raised by referee #1. Thanks for your time anyway.

**Responses to Referee #2**

We thank the referee #2 for the insightful comments and constructive suggestions. We have revised manuscript according to the reviewer's suggestions. Below are our responses to the referee's specific comments (*Italics*). The page and line numbers in our responses refer to those in the revised manuscript.

a. It is necessary to mention a summary of the authors' responses (a,b,c) in the text. Authors: Thanks for the reviewer's reminding, we have deleted the unused reference 'Adavi and Mashhadi' in the revised manuscript, and this revised manuscript has been proofread by a native English speaker.

b. The reference "Adavi and Mashhadi" is not used in the text. Please delete this reference from the list of references.

Authors: Thanks for the reviewer's reminding, we accepted your suggestions and deleted it in the revised manuscript.

c. Please consult with a native speaker on the language of the paper to complete the paper's corrections.

Authors: Thanks for the reviewer's suggestion, this revised manuscript has been proofread by a native English speaker.

**Responses to Referee #3**

We thank the referee #3 for the insightful comments and constructive suggestions. We have addressed all their comments in the revised manuscript. Below are our responses to the referee's specific comments (*Italics*). The page and line numbers in our responses refer to those in the revised manuscript.

*1 A brief introduction of the improved method needs to be included in the abstract.* Authors: Thanks for the reviewer's reminding, we accepted your suggestions and have added a brief introduction of the improved method into the abstract (Page 1, Lines 20-23).

The current English writing actually is not well. To be a high-quality journal paper, I would like to suggest author to polish the English by native speakers.

Authors: Thanks for the reviewer's suggestion, this paper has been polished by native English speakers.

The comparisons with the radiosonde show that the performance of the improved method is very poor. Especially in the lower troposphere, large differences exist. The poor performance will definitely restrict its applications in meteorology.

Authors: Thanks for the reviewer's reminding, to be frank we are not satisfied with the comparison results in the lower troposphere as you mentioned. However, we are really sorry to find that the large differences in the water vapor profiles occur due to the wrong calculation. In the revised manuscript, all calculations have been checked to ensure the correctness and the accuracy of water vapor derived from two models. The comparison with radiosonde data has been revised (Page 16, Lines 424-426; Page 17, Lines 434-435). Besides, the Figure 6 has also revised and we find the differences in the lower troposphere have shrunk a lot, not as bad as before (Page 17, Lines 439-449).

The improved method consists of two parts: using the traditional method to get the initial water vapor fields; performing polynomial fitting for each layer. In my opinion, the polynomial fitting is just another kind of horizontal constraint. Why not directly impose that horizontal constraint to the tomography equation? I am afraid if you perform polynomial fitting for each layer, the finally reconstructed SWVs will differ the measured SWVs a lot. Assessments on the reconstructed SWVs are required.

Authors: Thanks for the reviewer's question, we really appreciate your suggestion of directly imposing the layered optimal polynomial fitting functions as horizontal constraints to the tomography equation. In this paper, unfortunately, we did not think of this method. In our future research we would try this method to see if the results derived from the mentioned method would be better. As for the final reconstructed SWVs, the assessments on that are performed in the revised paper and the results show that the two model have the similar accuracy (Page 17-18, Lines 450-461). Referring to the previous paper we published (Yao and Zhao, 2017), this accuracy of SWVs comparison would be okay.

[revised manuscript text omitted]

for the experimental period (Unit: g/m3).